# Picocyanobacterial Contribution to the Total Primary Production in the Northwestern Pacific Ocean

Ho-Won Lee [1], Jae-Hoon Noh [1], Dong-Han Choi [1], Misun Yun [2], P. S. Bhavya [3], Jae-Joong Kang [4], Jae-Hyung Lee [5], Kwan-Woo Kim [4], Hyo-Keun Jang [4] and Sang-Heon Lee [4,*]

1   Marine Ecosystem Research Center, Korea Institute of Ocean Science and Technology, Busan 49111, Korea; howonlee@kiost.ac.kr (H.-W.L.); jhnoh@kiost.ac.kr (J.-H.N.); dhchoi@kiost.ac.kr (D.-H.C.)
2   College of Marine and Environmental Sciences, Tianjin University of Science and Technology, Tianjin 300457, China; misunyun@tust.edu.cn
3   Department of Biological Oceanography, Leibniz Institute for Baltic Sea Research Warnemuende, 18119 Rostock, Germany; bhavya.panthalil@io-warnemuende.de
4   Department of Oceanography, Pusan National University, Busan 46241, Korea; jaejung@pusan.ac.kr (J.-J.K.); goanwoo7@pusan.ac.kr (K.-W.K.); janghk@pusan.ac.kr (H.-K.J.)
5   South Sea Fisheries Research Institute, National Institute of Fisheries Science, Yeosu-si 59780, Korea; jhlee88@korea.kr
*   Correspondence: sanglee@pusan.ac.kr

**Abstract:** Picocyanobacteria (*Prochlorococcus* and *Synechococcus)* play an important role in primary production and biogeochemical cycles in the subtropical and tropical Pacific Ocean, but little biological information on them is currently available in the North Pacific Ocean (NPO). The present study aimed to determine the picocyanobacterial contributions to the total primary production in the regions in the NPO using a combination of a dual stable isotope method and metabolic inhibitor. In terms of cell abundance, *Prochlorococcus* were mostly dominant (95.7 $\pm$ 1.4%) in the tropical Pacific region (hereafter, TP), whereas *Synechococcus* accounted for 50.8%–93.5% in the subtropical and temperate Pacific region (hereafter, SP). Regionally, the averages of primary production and picocyanobacterial contributions were 11.66 mg C m$^{-2}$·h$^{-1}$ and 45.2% ($\pm$4.8%) in the TP and 22.83 mg C m$^{-2}$·h$^{-1}$ and 70.2% in the SP, respectively. In comparison to the carbon, the average total nitrogen uptake rates and picocyanobacterial contributions were 10.11 mg N m$^{-2}$·h$^{-1}$ and 90.2% ($\pm$5.3%) in the TP and 4.12 mg N m$^{-2}$·h$^{-1}$ and 63.5%, respectively. These results indicate that picocyanobacteria is responsible for a large portion of the total primary production in the region, with higher contribution to nitrogen uptake rate than carbon. A long-term monitoring on the picocyanobacterial variability and contributions to primary production should be implemented under the global warming scenario with increasing ecological roles of picocyanobacteria.

**Keywords:** cyanobacteria; *Prochlorococcus*; *Synechococcus*; primary production; northwestern Pacific Ocean

## 1. Introduction

Phytoplankton are major biological components as primary producers in marine ecosystems. Marine phytoplankton not only account for a significant proportion of global primary production, but also are an important food source in marine ecosystems and a potential moderator of global carbon cycle at the ocean–atmosphere interface [1]. Distribution, abundance, and diversity of phytoplankton differ greatly among dominant water masses in the various oceanic regions, which are closely related to physiochemical properties. In addition, long-term research on the limiting factors (e.g., temperature, nutrients, and light regime) of phytoplankton has reported that biological and ecological changes resulted from variations of these factors such as increasing of seawater temperature and reinforcement of stratification [2,3]. Primary production is widely used as one of key biological factors for

understanding the regional differences in basic environmental and biological conditions such as thermohaline properties, nutrients, chlorophyll-*a*, etc. [4–8].

The Pacific Ocean, due to its vastness extending from tropical regions to both the boundaries of polar oceans, is subjected to have distinctive climatic conditions at its various regions [4]. In the northwestern Pacific Ocean (NPO), the physico-chemical conditions are mainly influenced by North Equatorial Current, Kuroshio Current, Tsushima Warm Current, and pelagic/coastal water intrusions at the coastal zones in the East China Sea (ECS). In terms of phytoplankton community, autotrophic picoplankton communities were more abundant in the NPO than large-sized phytoplankton and heterotrophic bacteria [9,10]. Lee et al. [10] reported that autotrophic plankton (mainly pico-sized phytoplankton) comprised up to 80% of the total phytoplankton biomass in the euphotic zone, whereas the contribution of heterotrophic bacteria was 6–21% of phytoplankton biomass in the NPO. Furthermore, a few research works reported that picoplankton including pico-sized cyanobacteria (*Prochlorococcus* and *Synechococcus*) have been a significant component of biomass and primary production in the subtropical and tropical Pacific Ocean [11–17].

In general, *Prochlorococcus* exhibits a wide adaptation for the variability in light or nutrient conditions, whereas they are often observed to be limited by high temperature in the water column [16,18–23]. Other cyanobacterial species such as *Synechococcus* have relatively eurythermal characteristics and extend to low salinity waters. Hence, *Synechococcus* are widely distributed around the world ocean from tropical to polar waters with a high biomass in the upper euphotic zone [22,24]. Recently, it was reported that abundance and distribution of the small-sized autotrophic plankton communities including cyanobacteria, *Prochlorococcus,* and *Synechococcus*, increase in various oceans with global warming, which indicates that this issue is not limited to a local scale anymore [25,26].

Normally, the primary production by picophytoplankton (i.e., picocyanobacteria and picoeukaryotes) is estimated through filter fractionation [7,27–31]. However, it is difficult to distinguish carbon and/or nitrogen uptake rates between picocyanobacteria (*Prochlorococcus* and *Synechococcus*) and picoeukaryotes. Moreover, the fractionation in natural samples makes it difficult to physically separate picophytoplankton from heterotrophic bacteria, in case of nitrogen uptake [32]. Previous studies used metabolic inhibitors to partition the relative contributions of eukaryotes and prokaryotes in marine systems [33–35]. For example, cycloheximide inhibits the function of the 80-S ribosome of eukaryotes [36], whereas streptomycin specifically inhibits protein synthesis on the 70-S ribosome in bacteria [37]. Thus, these metabolic inhibitors could be effective in separating target organisms from non-target organisms [32]. Middelburg and Nieuwenhuize [34,35] successfully partitioned autotrophic and heterotrophic activity using metabolic inhibitors. Fouilland et al. [32] also applied metabolic inhibitors to partition the uptakes of nitrate, ammonium, and urea between prokaryotic and eukaryotic phytoplankton. As a result, they quantitatively reported the contribution of heterotrophic bacteria to nitrogen uptake [32].

In present study, a metabolic inhibitor (cycloheximide) based on the method of Fouilland et al. [32] was applied to measure picocyanobacterial contribution to the total primary production, since the inhibitor can remove the eukaryotes and directly determine only carbon and nitrogen uptake rates by picocyanobacteria in the samples. The objectives of this study were as follows: (1) to determine carbon and nitrogen uptake rates by picocyanobacteria and (2) to evaluate picocyanobacterial contribution to the primary production in the regions (subtropical-temperate Pacific region and tropical Pacific region) in the NPO.

## 2. Materials and Methods

### 2.1. Study Area and Sample Collection

The present study was conducted at 9 stations in the NPO during the POSEIDON cruise (13 May–4 June 2014) (Figure 1). In order to understand characteristics of primary productivity under the different environmental conditions in the NPO, our productivity stations were divided into two regions; the subtropical and temperate Pacific region (SP; A89 and A50), mainly affected by the coastal water of the ECS and tropical Pacific region

(TP; F10, F06, F03, F01, P03, P07, and P11), which are influenced by the Tsushima Warm Current, North Equatorial Current, and Kuroshio Current. Seawater samples from water column up to 1% light depths were collected using 10 L Niskin sampling bottles on the R/V Onnuri of the Korea Institute of Ocean Science and Technology (KIOST, Busan, Korea). The physical parameters (temperature and salinity) were determined using a Sea-Bird 911plus system (Sea-Bird, Inc., Brooklyn, NY, USA).

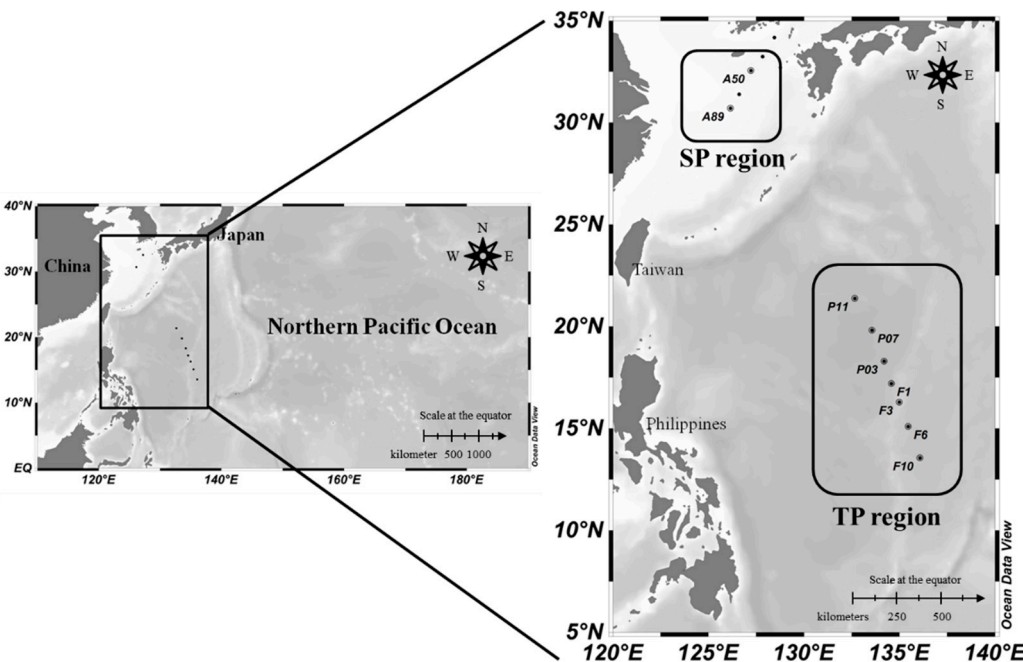

**Figure 1.** Sampling station in the two sampled regions of the northwestern Pacific Ocean; TP: tropical Pacific, SP: subtropical and temperate Pacific.

### 2.2. Measurements for Biomass and Abundance of Phytoplankton and Nutrient Concentrations

Chlorophyll-*a* (Chl-*a*) and phytoplankton abundance, as well as nutrient concentrations were measured at the 9 productivity stations. One liter of seawater for Chl-*a* concentrations presenting for phytoplankton biomass was filtered onto 25 mm GF/F filters. The filters were stored in a deep freezer and extracted within a month using 6 mL of 95% acetone by the method of Parsons et al. [38]. The final extracts were analyzed using a 10 AU fluorometer (Turner Design Inc., San Jose, CA, USA). Seawater samples for the enumeration and identification of major pico-sized phytoplankton groups (<2 μm) were counted by flow cytometry (BD Accury C6, BD Biosciences Inc., Mountain View, CA, USA) after staining with mixture of yellow-green and UV beads by the method of Olson et al. [39]. Nutrient data were provided by KIOST based on the standard colorimetric procedure [38].

### 2.3. Carbon and Nitrogen Uptake Rate Measurements

Total carbon and nitrogen uptake rates were measured at the 9 different stations using a $^{13}$C-$^{15}$N dual isotope tracer technique that has been applied in various oceans [27,40–43]. Seawater samples at 6 light depths (100%, 50%, 30%, 12%, 5%, and 1% of light intensity at surface) were collected from Niskin samplers to 1 L polycarbonate bottles covered with different LEE film screens (LEE Filters, Inc., Hampsire, UK) that corresponded to the different light levels. Further, the water samples were injected with enriched solutions of $^{13}$C (NaH$^{13}$CO$_3$) and $^{15}$N (K$^{15}$NO$_3$ or $^{15}$NH$_4$Cl) (less than 10% of the ambient concentrations) followed by deck incubation for 4 h. Hourly picocyanobacterial carbon and nitrogen uptake rates were measured at all the stations except station A50 in the SP using the dual isotope technique. For measuring the picocyanobacterial carbon and nitrogen uptake rates, the autotrophic eukaryotes were inhibited by a metabolic inhibitor (cycloheximide), which blocks

the cytoplasmic protein biosynthesis in 80-S ribosome of phytoplankton (eukaryotes) [32]. All the bottles were incubated in deck incubators along with primary productivity sample bottles for 4 h.

After incubation, seawater samples (0.5 L) for the carbon and nitrogen uptake rates were filtered onto the pre-combusted 25 mm GF/F filters. The filters were immediately frozen in the deep freezer until the analysis. Prior to the mass spectrometric analysis, samples were thawed, dried overnight, and packed in tin capsules. Particulate organic carbon (POC)/nitrogen (PON) and the amount of $^{13}C$ and $^{15}N$ were determined by Finnigan Delta + XL mass spectrometer at the Stable Isotope Facility, University of Alaska Fairbanks (UAF), USA after HCl fuming during 24 h for removing carbonate. Samples of analyzed total carbon and nitrogen uptake rates were calculated by using the methods of Hama et al. [44] and Dugdale and Goering [45]. Dark carbon uptake rates were subtracted for considering the heterotrophic bacterial process [46]. Because the carbon uptake rates from dark bottles were subtracted from the light bottles for removal of heterotrophic productivity without light, we assumed that only the contributions of autotrophic bacterial (i.e., picocyanobacterial) communities were obtained for the primary productivity.

## 3. Results

### 3.1. Physiochemical Structures in Water Column

Vertical profiles of temperature and salinity from all the stations in the NPO are shown in Figure 2. Surface temperature and salinity at the stations in the TP were higher than those in the SP. The average temperature and salinity in the upper water column were 17.3 °C and 33.2 psu in the SP, respectively, whereas they were 29.1 °C (S.D. = ±0.92 °C) and 34.8 psu (S.D. = ±0.52 psu) for TP, respectively.

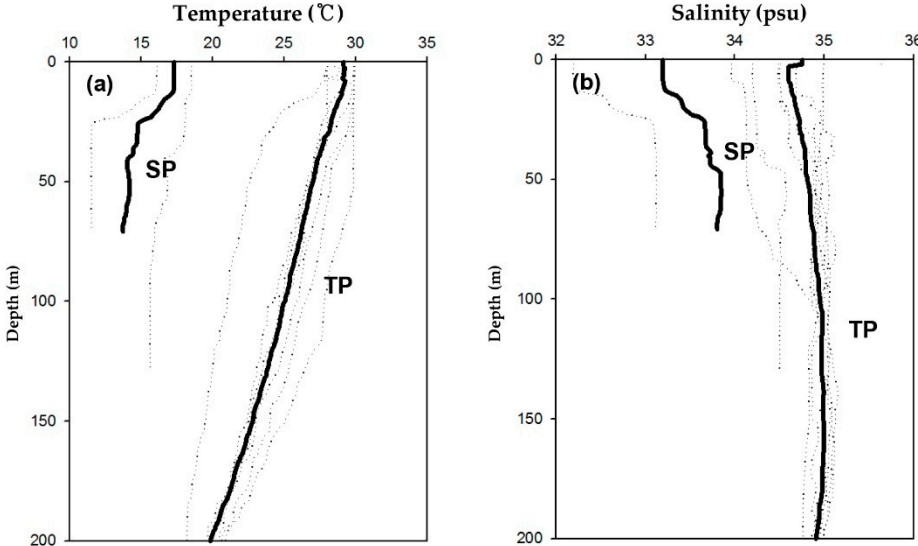

**Figure 2.** Vertical profiles of temperature (**a**), salinity (**b**) in the northwestern Pacific Ocean. Solid line represents average temperature and salinity in the TP and SP regions, respectively.

Generally, nutrient concentrations were depleted in both TP and SP regions except for 1% light depths (Figure 3). The mean nitrate concentrations within the euphotic zone were 0.13 (S.D. = ±0.35 μM) and 0.84 μM (S.D. = ±1.80 μM) in the TP and the SP, respectively. Ammonium concentrations were consistently low at euphotic zones of all the stations. The mean ammonium concentrations in the TP and the SP were 0.14 (S.D. = ±0.07 μM) and 0.18 μM (S.D. = ±0.03 μM), respectively. The euphotic depths at the stations in the TP were deeper than those in the SP (*t*-test, $p < 0.05$). The mean euphotic depths were 127.4 m (S.D. = ±16.5 m) and 35.0 m in the TP and the SP, respectively (Table 1).

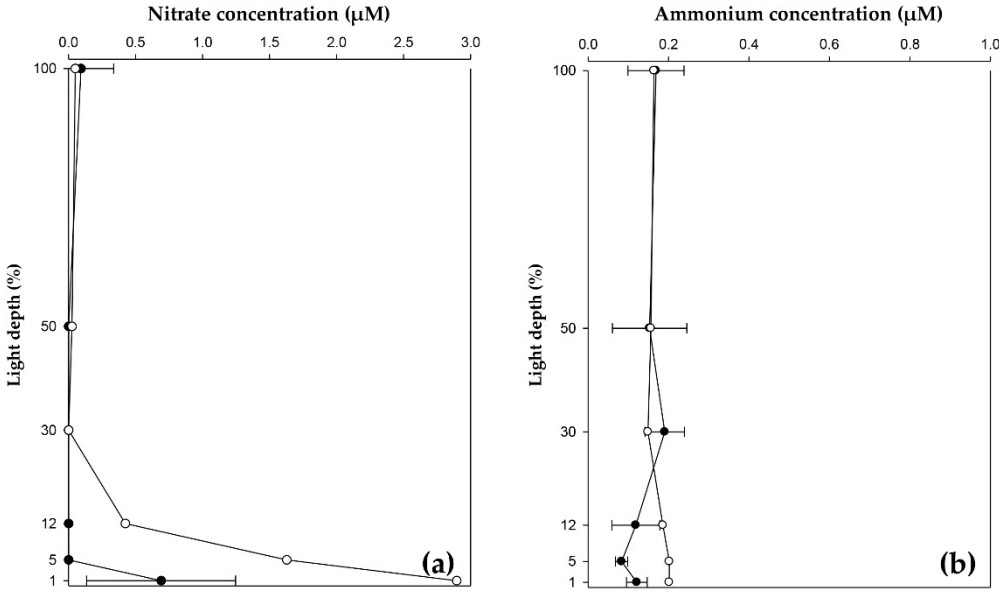

**Figure 3.** Vertical profiles of nitrate (**a**) and ammonium (**b**) concentrations averaged from each region (TP, closed circles; SP, open circles). SDs are shown by bars. Data courtesy of KIOST.

**Table 1.** The environmental conditions in the TP and SP regions of the northwestern Pacific Ocean.

|  | TP | | SP | |
|---|---|---|---|---|
|  | **Mean** | **S.D.** | **Mean** | **S.D.** |
| Temperature in the surface (°C) | 29.1 | 0.9 | 17.3 | - |
| Temperature in the euphotic depth (°C) | 26.7 | 2.3 | 15.3 | 2.3 |
| Salinity in the surface (psu) | 34.8 | 0.5 | 33.2 | - |
| Salinity in the euphotic zone (psu) | 34.8 | 0.6 | 33.9 | 0.6 |
| Nitrate in the euphotic zone (µM) | 0.13 | 0.35 | 0.84 | 1.8 |
| Ammonium in the euphotic depth (µM) | 0.14 | 0.07 | 0.18 | 0.03 |
| Euphotic depth (m) | 127.4 | 16.5 | 35 | - |

### 3.2. Distribution of Phytoplankton in Water Column

The average euphotic depth-integral total Chl-*a* concentrations were 15.0 (S.D. = ±6.6 mg Chl-*a* m$^{-2}$) and 18.1 mg Chl-*a* m$^{-2}$ in the TP and SP, respectively (Table 2). Although the integral total Chl-*a* concentrations were not significantly different between the TP and SP locations (Table 2), the vertical distributions of Chl-*a* were obviously different between the two locations (Figure 4). Deep chlorophyll maximum (DCM) layers, in which the Chl-*a* concentrations were significantly (*t*-test, $p < 0.01$) higher compared to those at the surface, were observed at the bottom (1% light depth) of the euphotic zone in the TP. However, no substantial DCM layers were found in the SP (Figure 4).

The cell abundance of autotrophic plankton, including picocyanobacteria (*Synechococcus* and *Prochlorococcus*) and picoeukaryotes, were different between the TP and the SP (Figure 5). The average depth-integral abundance *of Synechococcus*, *Prochlorococcus*, and picoeukaryotes in the TP were $1.85 \times 10^{11}$ (S.D. = $\pm 0.64 \times 10^{11}$ cells m$^{-2}$), $0.64 \times 10^{13}$ (S.D. = $\pm 0.10 \times 10^{13}$ cells m$^{-2}$), and $0.96 \times 10^{11}$ cells m$^{-2}$ (S.D. = $\pm 0.31 \times 10^{11}$ cells m$^{-2}$), respectively (Figure 5). In the SP, the cell abundance of *Synechococcus* and picoeukaryotes were $14.4 \times 10^{11}$ and $4.28 \times 10^{11}$ cells m$^{-2}$, respectively. No *Prochlorococcus* were generally found in the SP except some at 46 m of A89 (Figure 5). Consequently, *Prochlorococcus* (mean ± S.D. = 95.7 ± 1.4%), *Synechococcus* (mean ± S.D. = 2.8 ± 1.0%), and picoeukaryotes (mean ± S.D. = 1.4 ± 0.4%) contributed the plankton community in the TP. In contrast, *Synechococcus* accounted for 93.5% and 50.8%, whereas picoeukaryotes were 5.2% and 49.2% at A89 and A50 in the SP, respectively.

**Table 2.** Chlorophyll-*a*, C/N ratio, *f*-ratio, carbon, and nitrogen (nitrate and ammonium) uptake rates by total phytoplankton communities in the TP and SP regions of the northwestern Pacific Ocean.

| | TP | | | SP | | |
|---|---|---|---|---|---|---|
| | **Mean** | **S.D.** | ***n*** | **Mean** | **S.D.** | ***n*** |
| Integrated total Chlorophyll-a (mg Chl-*a* m$^{-2}$) | 15 | 6.6 | 7 | 18.1 | - | 2 |
| C/N ratio (atom/atom) | 11 | 1.8 | 7 | 9.8 | - | 2 |
| Carbon specific uptake (h$^{-1}$) | 0.001508 | 0.001034 | 42 | 0.004951 | 0.004069 | 12 |
| Carbon absolute uptake (mg C m$^{-3}\cdot$h$^{-1}$) | 0.099 | 0.068 | 42 | 0.688 | 0.653 | 12 |
| Integrated carbon uptake (mg C m$^{-2}\cdot$h$^{-1}$) | 11.66 | 4.8 | 7 | 20.85 | - | 2 |
| Nitrate specific uptake (h$^{-1}$) | 0.000632 | 0.000435 | 42 | 0.001097 | 0.001096 | 12 |
| Nitrate absolute uptake (mg NO$_3^-$ m$^{-3}\cdot$h$^{-1}$) | 0.007987 | 0.006853 | 42 | 0.022084 | 0.024058 | 12 |
| Integrated nitrate uptake (mg NO$_3^-$ m$^{-2}\cdot$h$^{-1}$) | 1.06 | 0.68 | 7 | 0.69 | - | 2 |
| Ammonium specific uptake (h$^{-1}$) | 0.006756 | 0.003664 | 42 | 0.006355 | 0.003179 | 12 |
| Ammonium absolute uptake (mg NH$_4^+$ m$^{-3}\cdot$h$^{-1}$) | 0.072252 | 0.044304 | 42 | 0.120235 | 0.077651 | 12 |
| Integrated ammonium uptake (mg NH$_4^+$ m$^{-2}\cdot$h$^{-1}$) | 9.05 | 3.1 | 7 | 4.05 | - | 2 |
| Nitrogen specific uptake (h$^{-1}$) | 0.007388 | 0.004099 | 42 | 0.007452 | 0.004255 | 12 |
| Nitrogen absolute uptake (mg N m$^{-2}\cdot$h$^{-1}$) | 0.08 | 0.047 | 42 | 0.142 | 0.1 | 12 |
| Integrated nitrogen uptake (mg N m$^{-2}\cdot$h$^{-1}$) | 10.11 | 2.49 | 7 | 4.74 | - | 2 |
| *f*-ratio | 0.1 | 0.03 | 7 | 0.13 | - | 2 |

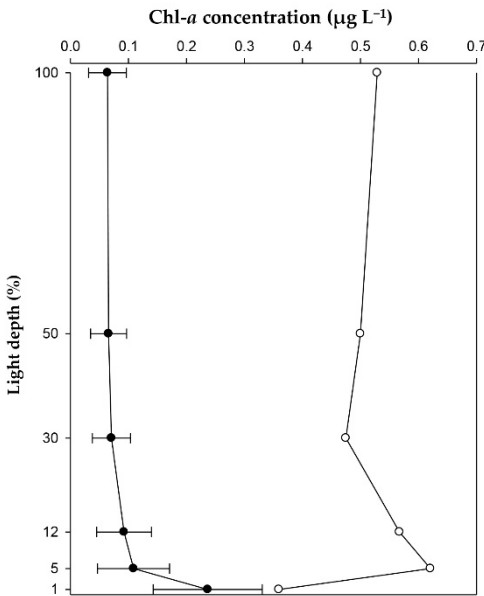

**Figure 4.** Vertical profiles of chlorophyll-*a* concentrations in the TP (closed circles) and SP (open circles) regions. SDs are shown by bars.

### 3.3. Total Carbon and Nitrogen Uptake Rates in the NPO

The largest carbon uptake rate was at 100% light depth at each station in the SP, whereas in the TP, the largest rate was observed at 30–50% light depths (Figure 6a). The lowest carbon uptake rate was found at the chlorophyll-maximum layer corresponding to 1% light depth in the SP. The average rates of carbon uptake at each light depth were significantly higher in the SP (*t*-test, $p < 0.05$) than in the TP. The ranges of depth-integrated carbon uptake rates in the TP and SP were 3.29–16.89 mg C m$^{-2}\cdot$h$^{-1}$ with an average of 11.66 mg C m$^{-2}\cdot$h$^{-1}$ and 9.17–32.54 mg C m$^{-2}\cdot$h$^{-1}$ with an average of 20.85 mg C m$^{-2}\cdot$h$^{-1}$, respectively (Figure 7a and Table 2). Based on our dark carbon uptake rates in this study, the heterotrophic contributions to the total primary productions were 1.5% (S.D. = ±0.7%) and 8.7% (S.D. = ±12.8%) for the SP and the TP, respectively.

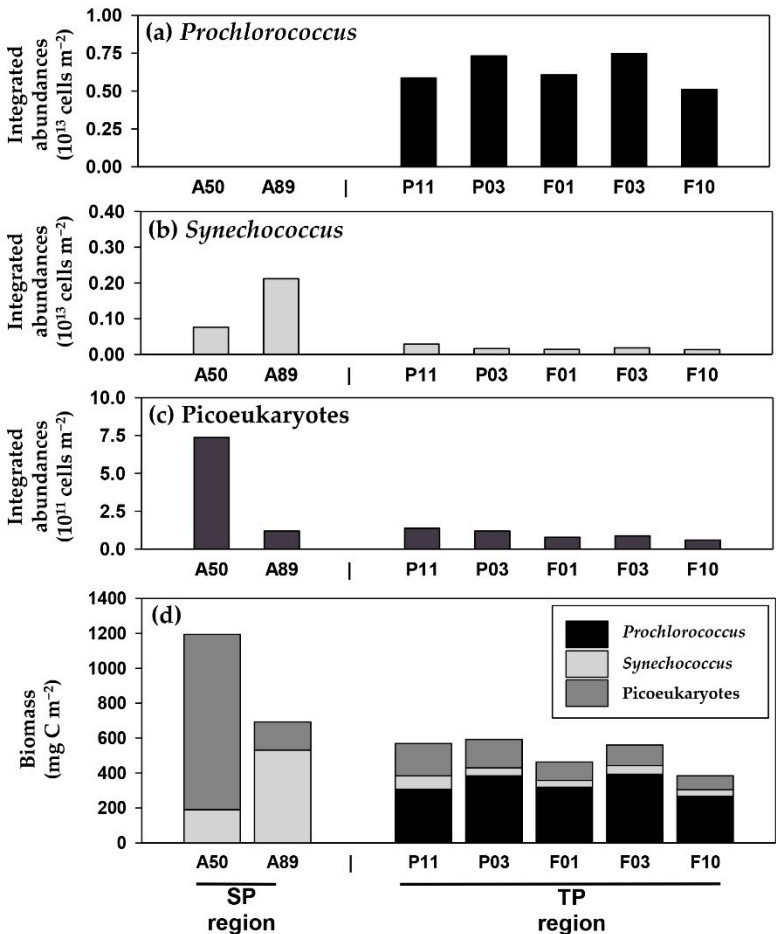

**Figure 5.** Integrated abundance of *Prochlororoccus* (**a**), *Synechococcus* (**b**), and picoeukaryotes (**c**) in the euphotic zone. Biomass of *Prochlororoccus*, *Synechococcus*, and picoeukaryotes in the euphotic zone (**d**) in the euphotic zone.

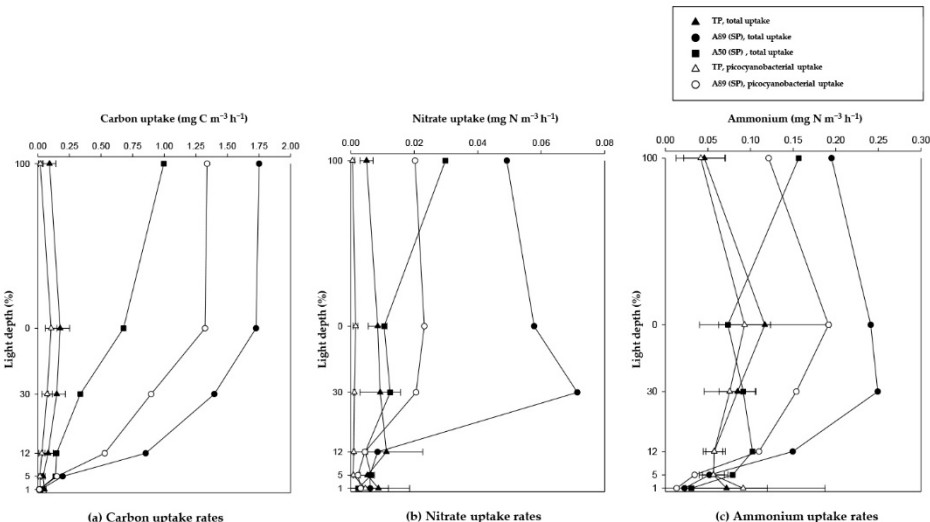

**Figure 6.** Vertical profiles of the total carbon and nitrogen absolute uptake rates (TP, closed triangles; A89, closed circles; A50, closed squares) and picocyanobacterial carbon and nitrogen absolute uptake rates (TP, open triangles; A89, open circles) in the TP and SP regions in the North Pacific Ocean. SDs are shown by bars. Carbon uptake rates (**a**), Nitrate uptake rates (**b**), and Ammonium uptake rates (**c**).

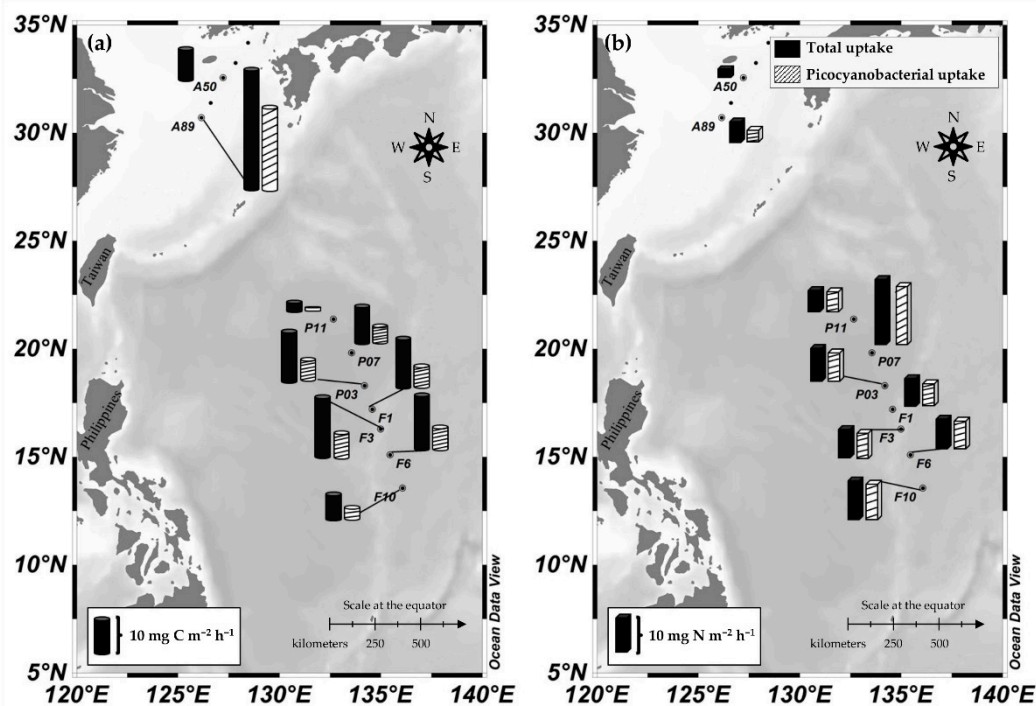

**Figure 7.** Regional distribution of total carbon and nitrogen uptake rates ((**a**), left) and picocyanobacterial carbon and nitrogen uptake rates ((**b**), right) in the northwestern Pacific Ocean. Bars with diagonal stripes indicate carbon and nitrogen uptake rates of picocyanobacterial communities.

Nitrogen uptake rates did not show any significant pattern with light depths as carbon uptake rates (Figure 6b,c). The depth-integrated nitrogen (nitrate+ammonium) uptake rates in the TP and SP ranged from 6.52 to 17.96 mg N m$^{-2}$·h$^{-1}$ with an average of 10.11 mg N m$^{-2}$·h$^{-1}$ and from 2.98 mg N m$^{-2}$·h$^{-1}$ to 6.50 mg N m$^{-2}$·h$^{-1}$ with an average of 4.74 mg N m$^{-2}$·h$^{-1}$, respectively (Figure 7b and Table 2). In detail, the mean of nitrate and ammonium uptake rates in the TP were 1.06 mg N m$^{-2}$·h$^{-1}$ and 9.05 mg N m$^{-2}$·h$^{-1}$, respectively, whereas those in the SP were 0.69 mg N m$^{-2}$·h$^{-1}$ and 4.05 mg N m$^{-2}$·h$^{-1}$, respectively. Ammonium uptake rates were substantially higher than nitrate uptake rates in both regions.

*3.4. Picocyanobacterial Carbon and Nitrogen Uptakes in the NPO*

The average rates of picocyanobacterial carbon uptakes showed similar trends like vertical abundance profiles of these predominant species (Figure 8). Vertical profiles of picocyanobacterial carbon, nitrate, and ammonium uptake rates showed similar trends as those of the uptake rates by total phytoplankton community at each light depth (Figure 6). Picocyanobacterial carbon uptake rates integrated from the euphotic depths were 5.31 mg C m$^{-2}$·h$^{-1}$ (S.D. = ±2.16 mg C m$^{-2}$·h$^{-1}$) in the TP, whereas the integrated carbon uptake rates by picocyanobacteria at the A89 (SP) was 22.8 mg C m$^{-2}$·h$^{-1}$ (Figure 9a). The average rates of picocyanobacterial carbon uptake at each light gradient were significantly higher in the SP (Table 3; *t*-test, *p* < 0.05). Integrated hourly picocyanobacterial nitrogen uptake rates were 6.32–16.16 mg N m$^{-2}$·h$^{-1}$ with an average of 9.10 mg N m$^{-2}$·h$^{-1}$ in the TP and 4.12 mg N m$^{-2}$·h$^{-1}$ at the A89 in the SP (Figures 7b and 9b). The average nitrate uptake rates by picocyanobacterial communities in the TP and A89 (SP) were 0.21 mg N m$^{-2}$·h$^{-1}$ (S.D. = ±0.20 mg N m$^{-2}$·h$^{-1}$) and 0.40 mg N m$^{-2}$·h$^{-1}$, respectively, whereas the average ammonium uptake rates of picocyanobacterial communities were 8.89 mg N m$^{-2}$·h$^{-1}$ (S.D. = ±3.18 mg N m$^{-2}$·h$^{-1}$) and 3.72 mg N m$^{-2}$·h$^{-1}$, respectively (Table 3). Picocyanobacterial ammonium uptake rates were more than the nitrate uptake rates in the NPO (Figure 9c,d).

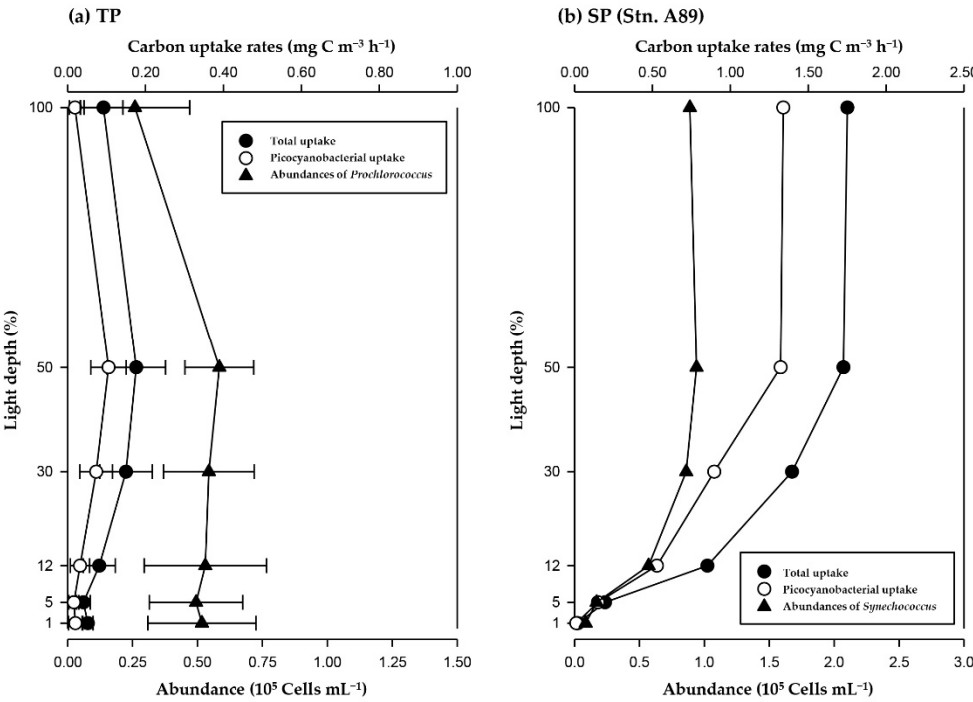

**Figure 8.** Comparison of the uptake rates for total (closed circles) and bacterial (open circles) carbon uptake with the abundances of the predominant species (closed triangles) in the northwestern Pacific Ocean. TP (**a**) and SP (**b**). SDs are shown by bars.

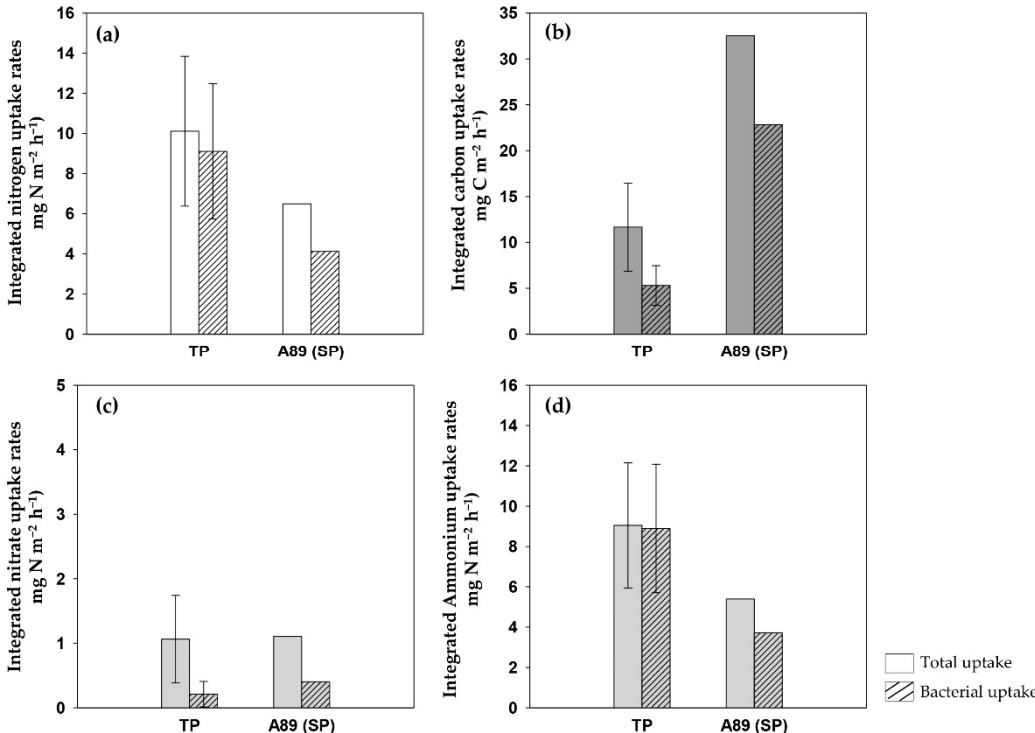

**Figure 9.** Picocyanobacterial contribution to total carbon and nitrogen uptake rates (primary productivity) in the TP and SP regions of the northwestern Pacific Ocean. Unicolor bars indicate total uptake of each uptake rate, whereas other bars with diagonal stripes indicate picocyanobacterial uptake rates. SDs are shown by bars. Integrated nitrogen uptake rates (**a**), Integrated carbon uptake rates (**b**), Integrated nitrate uptake rates (**c**), and Integrated ammonium uptake rates (**d**).

**Table 3.** Carbon and nitrogen (nitrate and ammonium) uptake rates by picocyanobacterial communities in the TP and SP regions of the northwestern Pacific Ocean.

| | TP | | | SP (A89) | | |
|---|---|---|---|---|---|---|
| | Mean | SD | $n$ | Mean | SD | $n$ |
| Picocyanobacterial carbon specific uptake ($h^{-1}$) | 0.000695 | 0.000548 | 42 | 0.004404 | 0.003065 | 6 |
| Picocyanobacterial carbon absolute uptake (mg C $m^{-3} \cdot h^{-1}$) | 0.044 | 0.043 | 42 | 0.708 | 0.573 | 6 |
| Integrated picocyanobacterial carbon uptake (mg C $m^{-2} \cdot h^{-1}$) | 5.31 | 2.16 | 7 | 22.83 | - | 1 |
| Picocyanobacterial nitrate specific uptake ($h^{-1}$) | 0.000104 | 0.000174 | 42 | 0.00047 | 0.000348 | 6 |
| Picocyanobacterial nitrate absolute uptake (mg $NO_3^-$ $m^{-3} \cdot h^{-1}$) | 0.001577 | 0.003138 | 42 | 0.012296 | 0.00997 | 6 |
| Integrated picocyanobacterial nitrate uptake (mg $NO_3^-$ $m^{-2} \cdot h^{-1}$) | 0.21 | 0.2 | 7 | 0.4 | - | 1 |
| Picocyanobacterial ammonium specific uptake ($h^{-1}$) | 0.005073 | 0.002693 | 42 | 0.004497 | 0.002518 | 6 |
| Picocyanobacterial ammonium absolute uptake (mg $NH_4^+$ $m^{-3} \cdot h^{-1}$) | 0.070531 | 0.046278 | 42 | 0.104054 | 0.068611 | 6 |
| Integrated picocyanobacterial ammonium uptake (mg $NH_4^+$ $m^{-2} \cdot h^{-1}$) | 8.89 | 3.18 | 7 | 3.72 | - | 1 |
| Integrated picocyanobacterial nitrogen uptake (mg N $m^{-2} \cdot h^{-1}$) | 9.1 | 3.37 | 7 | 4.12 | - | 1 |
| Picocyanobacterial $f$-ratio | 0.02 | 0.01 | 7 | 0.1 | - | 1 |

## 4. Discussion

In this study, the abundance of picophytoplankton was different between the TP and the SP (Figure 5). *Prochlorococcus* were not found but *Synechococcus* and picoeukaryotes co-occurred in the SP, whereas *Prochlorococcus* were the dominant picophytoplankton population in the TP. The difference in abundance of dominant population observed in the TP and the SP might be due to different physico-chemical properties as the result of the major currents. Because distribution and abundance of phytoplankton in the euphotic zone can be altered by the hydrological conditions of the seawater, these physiochemical properties are determined by the major currents [47–50]. In fact, the TP is directly influenced by North Equatorial Current, whereas the SP is influenced mainly by the Kuroshio Current, Tsushima Warm Current, and coastal fresh water, respectively [16,51]. According to Choi et al. [24], the picocyanobacterial distribution in the NPO was distinguished along the physical and chemical properties of the water masses. In this study, the water depth in the SP was shallow and had lower temperature and salinity than the TP (Figure 2), whereas the TP was a typical high-temperature oligotrophic water. Since *Prochlorococcus* have been found to be more abundant in the oligotrophic conditions because of their ecological plasticity with respect to requirements of nutrients and light [18,52–55], *Prochlorococcus* could be dominant under temperature and oligotrophic TP. According to previous studies [12,55], *Synechococcus* are usually dominant in the mesotrophic seawater or shallow waters. Thus, *Synechococcus* and picoeukaryotes could be abundant in relatively mesotrophic and shallow SP, which is consistent with previous study from the western Pacific Ocean [54].

In terms of carbon biomasses estimated from the average carbon contents [56,57], *Prochlorococcus* contributed 66.1% to the total phytoplankton in the TP (Figure 5d). In the SP, *Synechococcus* were 76.4% at A89 and picoeukaryotes were 84.0% at A50, respectively. Especially, the carbon biomass contribution of picoeukaryotes was higher than that of *Synechococcus* at the A50, although picoeukaryotes had lower cell abundances than *Synechococcus*, because picoeukaryotes have higher carbon contents compared to *Synechococcus*.

Based on the hourly carbon uptake rates by total phytoplankton, which were estimated in this study, the average daily primary productivities were 0.15 g C $m^{-2} \cdot d^{-1}$ (S.D. = $\pm 0.06$ g C $m^{-2} \cdot d^{-1}$) and 0.29 g C $m^{-2} \cdot d^{-1}$ in the TP and SP, respectively (Table 4). Our daily primary productivities fell within the range of previous studies in both regions [4,5,51]. In

the TP, Taniguchi [4] reported 0.09 g C m$^{-2}$·d$^{-1}$ in the North Equatorial Current (Table 4). Kwak et al. [5] observed a relatively higher range of daily primary productivity from 0.17 to 0.23 g C m$^{-2}$·d$^{-1}$ in the western Pacific Ocean (Table 4). For the SP, the average daily primary productivity obtained from this study is comparable with those from other previous studies [5,51]. Gong et al. [51] reported 0.31 ± 0.16 g C m$^{-2}$·d$^{-1}$ and 0.52 ± 0.32 g C m$^{-2}$·d$^{-1}$ during early spring and summer, respectively (Table 4). Our daily primary productivity is nearly identical to the daily production (0.28 g C m$^{-2}$·d$^{-1}$) reported by Kwak et al. [5] (Table 4).

**Table 4.** Comparison of daily primary productivity with previous studies in the northwestern Pacific Ocean.

| Region | References | Carbon Uptake Rates | Nitrate Uptake Rates | Ammonium Uptake Rates | |
|---|---|---|---|---|---|
| | | Average ± SD (g C m$^{-2}$·d$^{-1}$) | Average ± SD (g N m$^{-2}$·d$^{-1}$) | Average ± SD (g N m$^{-2}$·d$^{-1}$) | Season |
| TP | Taniguchi (1972) | 0.09 | - | - | Winter |
| | Kwak et al. (2013) | 0.2 | - | - | Summer |
| | In this study | 0.15 ± 0.06 | 0.01 ± 0.01 | 0.16 ± 0.01 | Late spring |
| SP | Gong et al. (2003) | 0.31 ± 0.16 | - | - | Early spring |
| | | 0.52 ± 0.32 | - | - | Summer |
| | Kwak et al. (2013) | 0.28 | - | - | Summer |
| | In this study | 0.45 (A89) | 0.02 (A89) | 0.10 (A89) | Late spring |
| | | 0.13 (A50) | 0.01 (A50) | 0.05 (A50) | |

Daily total ammonium uptake rates were calculated by multiplying hourly nitrogen uptake rates and each photoperiod [58] in this study. The average daily total ammonium uptake rates were higher than total nitrate uptakes in the euphotic zone of both regions. The average daily total ammonium and nitrate uptake rates were 0.16 g N m$^{-2}$·d$^{-1}$ (S.D. = ±0.06 g N m$^{-2}$·d$^{-1}$) and 0.01 g N m$^{-2}$·d$^{-1}$ (S.D. = ±0.01 g N m$^{-2}$·d$^{-1}$) in the TP, respectively (Table 4). In the SP, the daily total ammonium and nitrate uptake rates were 0.07 g N·m$^{-2}$ d$^{-1}$ and 0.01 g N m$^{-2}$·d$^{-1}$ at A89, respectively (Table 4). Accordingly, average $f$-ratios ([nitrate uptake rate]/[nitrate+ammonium uptake rate], [59]) were 0.10 (S.D. = ±0.03) and 0.13 in the TP and SP (Table 2), respectively, as a result of prominent ammonium uptakes. This indicates that a main nitrogen source for growth of total autotrophic plankton was mainly supported by regenerated ammonium in this region during our observation period.

In this study, the average picocyanobacterial contributions to the total primary productivity accounted for 45.2% (S.D. = ±4.8%) in the TP and 70.2% in the A89 (SP) (Figure 9a). Glover et al. [12] reported that contribution of *Synechococcus* to the total primary production, which varies from 6% to 46% in different water masses in the northwestern Atlantic Ocean. In contrast, Liu et al. [15] observed a high contribution of *Prochlorococcus* up to 82% to the primary production at Station ALOHA in the subtropical North Pacific Ocean.

Based on each nitrate and ammonium uptake rate, the average picocyanobacterial $f$-ratios were 0.02 (S.D. = ±0.01) and 0.10 in the TP and A89 (SP), respectively (Table 3). This result suggests that picocyanobacterial communities strongly depended on a regenerated nitrogen source (i.e., ammonium) or N$_2$ fixation in our study area during the observation period. From the comparison of $f$-ratios between the total phytoplankton and picocyanobacterial communities, we found that picocyanobacterial $f$-ratios were substantially lower compared to those of the total phytoplankton communities in the two regions (Tables 2 and 3).

Depth integrated hourly nitrogen uptake rates of picocyanobacterial communities were 9.10 mg N m$^{-2}$·h$^{-1}$ (S.D. = ±3.73 mg N m$^{-2}$·h$^{-1}$) and 4.12 mg N m$^{-2}$·h$^{-1}$ in the TP and the A89 (SP), respectively (Figure 9). The total nitrogen uptake rates at the same regions were 10.11 mg N m$^{-2}$·h$^{-1}$ and 6.50 mg N m$^{-2}$·h$^{-1}$, respectively. Given the nitrogen uptake rates, the average picocyanobacterial contributions to the total nitrogen uptake rates were 90.2% (S.D. = ±5.3%) and 63.5% in the TP and the A89 (SP), respectively, in this study. These picocyanobacterial contributions to the total nitrogen uptake rates

are substantially higher than those to the total carbon uptake rates of the total plankton communities in TP. However, the nitrogen utilization by heterotrophic bacteria can be important since the heterotrophic bacteria account for a large fraction of nitrogen uptake in the global ocean including the Arctic Ocean [32,60,61]. Although we are incapable of distinguishing each contribution for nitrogen uptake between heterotrophic bacteria and picocyanobacteria from this study using a metabolic inhibitor (cycloheximide) blocking only photosynthetic eukaryotes, we need to consider the heterotrophic bacterial nitrogen utilization from the nitrogen contributions in future studies. Apart from this, the potential N2 fixation by cyanobacteria can vary with environmental conditions, particularly nutrient stoichiometry [62]. When the $NH_4^+$ concentration is relatively higher than phosphorous, the nitrogenase activity can be stopped and photosynthesis can be activated. On the other hand, if the $NH_4^+$:P ratio is lower than the Redfield's ratio, N2 fixation can be a more major process than primary production. So, the contribution of picocyanobacteria towards the total primary production can be underestimated in that case. Furthermore, when autotrophic primary production is stopped by the inhibitor, the competition for nutrients in the samples may be lesser than one with autotrophic activity and, hence, the primary production rates by picocyanobacteria could be overestimated. Currently, there are some uncertainties for estimating picocyanobacterial contributions to the primary production and nitrogen uptake rates. Therefore, the combined approaches using several different applications are highly important for further future studies on cyanobacterial ecological roles in various oceans.

## 5. Summary and Conclusions

In this study, we measured picocyanobacterial contribution to the carbon and nitrogen uptake rates by total phytoplankton in the regions of the NPO. There are different abundances and biomasses of dominant species in the TP and the SP regions. *Prochlorococcus* and *Synechococcus* were abundant in the TP and the SP regions, respectively. The picocyanobacterial contributed 45.2% (S.D. = ±4.8%) to primary production by total picophytoplankton in the TP, whereas the picocyanobacterial contribution was about 70.2% in the SP. The picocyanobacterial community is believed to be more important in terms of nitrogen uptake rates since they could contribute about 90.2% (S.D. = ±5.3%) to the total nitrogen uptake rates by picophytoplankton in both regions.

The importance of picoplankton including cyanobacteria has been raised continuously in research regarding the global ocean [25,63,64]. In particular, the picocyanobacterial *Prochlorococcus* and *Synechococcus* have significant ecological positions in the biomass and production in the ocean, but the relative contributions of these organisms to primary productivity are different under various environmental conditions [22]. Under the global warming scenario, picoplankton contribution relative to large plankton will increase in the strongly stratified upper ocean [3]. This climate change will result in increasing distribution, abundance, and contributions to primary production of picocyanobacteria, especially in tropical and subtropical oceans and, consequently, will cause large impacts on the global ocean ecosystem and biogeochemical cycles [26]. Therefore, more measurements under various environmental conditions are needed to better understand the role of picocyanobacterial in the ecosystem.

**Author Contributions:** Conceptualization, H.-W.L., J.-H.N. and S.-H.L.; methodology, H.-W.L., P.S.B. and S.-H.L.; validation, H.-W.L., J.-H.N. and S.-H.L.; formal analysis, H.-W.L., J.-H.N., D.-H.C., J.-J.K., J.-H.L. and K.-W.K.; investigation, H.-W.L., J.-H.N. and D.-H.C.; writing—original draft preparation, H.-W.L.; writing—review and editing, M.Y., P.S.B. and S.-H.L.; visualization, J.-J.K., J.-H.L., K.-W.K. and H.-K.J.; supervision, S.-H.L. All authors have read and agreed to the published version of the manuscript.

**Funding:** This research was supported by the National Research Foundation of Korea (NRF) grant funded by the Korean government (MSIT; NRF-2019R1A2C1003515) and partly by the Korea Institute of Ocean Science and Technology (KIOST; PE99923).

**Institutional Review Board Statement:** Not applicable.

**Informed Consent Statement:** Not applicable.

**Data Availability Statement:** Not applicable.

**Acknowledgments:** We thank the captain and crew members of the R/V *Onnuri* for their outstanding assistance. Especially, we very much appreciate the KIOST for providing CTD and nutrient data. Finally, we thank the anonymous reviewers who greatly improved an earlier version of manuscript.

**Conflicts of Interest:** The authors declare no conflict of interest.

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
