# Peer review of "Picocyanobacterial Contribution to the Total Primary Production in the Northwestern Pacific Ocean"

_water, doi:10.3390/w13111610_

Round 1

Reviewer 1 Report

Review Report for Water 1205224

This is a very well-written paper. The authors put in a lot of effort in data collection, analysis, and writing. I have very minor comments, especially with respect to Table 4.

  1. Maybe I missed some of the explanations with respect to Table 4. Since the other studies by Gong and Kwak were carried out in early Spring and Summer, the present study was carried out in late Spring. Is this deliberate? What I meant is that the authors deliberately performed a study different from other people so that the new study can offer some complementary observations. Is my interpretation correct?
  2. If my interpretation above is incorrect, then the comparisons in Table 4 may not be meaningful. This is because different seasons will have different uptake rates due to seasonal variations. Please include some additional explanations for Table 4.
  3. The sampling stations require a lot of manpower to maintain and manage. I am curious about whether or not some remote sensing methods could be used for achieving the same objective. Nowadays, there are many satellite imagers such as Landsat, Worldview, etc. Can anyone of those imagers offer some help?

Author Response

This is a very well-written paper. The authors put in a lot of effort in data collection, analysis, and writing. I have very minor comments, especially with respect to Table 4.

  1. Maybe I missed some of the explanations with respect to Table 4. Since the other studies by Gong and Kwak were carried out in early Spring and Summer, the present study was carried out in late Spring. Is this deliberate? What I meant is that the authors deliberately performed a study different from other people so that the new study can offer some complementary observations. Is my interpretation correct?

==>Yes, it is correct. We would like to compare seasonal daily primary productivity in the region. On top of that, we’d like to investigate the contribution of picophytoplankton to the total productivity in this study.

  1. If my interpretation above is incorrect, then the comparisons in Table 4 may not be meaningful. This is because different seasons will have different uptake rates due to seasonal variations. Please include some additional explanations for Table 4.

==> Please see the comments for # 1!

  1. The sampling stations require a lot of manpower to maintain and manage. I am curious about whether or not some remote sensing methods could be used for achieving the same objective. Nowadays, there are many satellite imagers such as Landsat, Worldview, etc. Can anyone of those imagers offer some help?

==> Yes for the primary productivity (or, carbon uptake rate) using satellite methods! However, not for the nitrate and ammonium uptake rates. Especially, in this study, our main purpose is to investigate the contribution of picophytoplankton especially for picocyanobacteria to the productivity, which is not possible using satellite approaches at current stage.

Reviewer 2 Report

The article is devoted to an important issue, the assessment of the participation of picoplankton autotrophs in the creation of the primary production of the Pacific Ocean. The authors analyzed the available work and methods for this assessment. The simplest and most technologically advanced methods were chosen to establish the proportion of participation of each of the groups of organisms, Protochlorococcus and Synechococcus from Cyanobacteria, as well as undifferentiated picoeukaryotes. The results are clear and statistically sufficiently processed. The conclusions correspond to the tasks set. The article can be published in the journal WATER with minor changes. The attached file contains remarks of a more morphological nature. However, one point needs to be clarified in more detail. Since the authors' conclusions concern the difference in the contribution of the three studied groups of picoplankton autotrophs to total primary production, it is necessary to understand how the collected organisms were determined to the generic level. It is clear that a cytometer was used, but how the authors not only calculated, but also identified the mentioned species.

Author Response

The article is devoted to an important issue, the assessment of the participation of picoplankton autotrophs in the creation of the primary production of the Pacific Ocean. The authors analyzed the available work and methods for this assessment. The simplest and most technologically advanced methods were chosen to establish the proportion of participation of each of the groups of organisms, Protochlorococcus and Synechococcus from Cyanobacteria, as well as undifferentiated picoeukaryotes. The results are clear and statistically sufficiently processed. The conclusions correspond to the tasks set. The article can be published in the journal WATER with minor changes. The attached file contains remarks of a more morphological nature.

However, one point needs to be clarified in more detail. Since the authors' conclusions concern the difference in the contribution of the three studied groups of picoplankton autotrophs to total primary production, it is necessary to understand how the collected organisms were determined to the generic level. It is clear that a cytometer was used, but how the authors not only calculated, but also identified the mentioned species.

==> Yes, we used a flow cytometer approach to determine the numbers and species by cell sizes and different light scattering characteristics for each species based on the methods from Olson et al. (2003) and Marie et al. (2005). We briefly mentioned this in our revised manuscript in line 117-118, page 3.

References

Olson, R.J.; Shalapyonok, A.; Sosik, H.M. An automated submersible flow cytometer for analyzing pico- and nanophytoplankton: FlowCytobot. Deep. Res. Part I Oceanogr. Res. Pap. 2003, 50, 301–315

Marie et al. (2005)
Marie, D.; Simon, N.; Vaulot, D.; Phytoplankton cell counting by flow cytometry. Algal Culture Techniques, 2005, 253-267, DOI: 10.1016/B978-012088426-1/50018-4

Reviewer 3 Report

Summary:

In this manuscript, the authors attempt to quantify the contributions of picocyanobacteria (specifically the genera Prochlorococcus and Synechococcus) to the total primary production in the North Pacific Ocean. The authors employed flow cytometry to estimate the abundance of the picocyanobacteria and stable isotope mass spectrometry to estimate the Carbon and Nitrogen uptake rates. They have reported that the abundance of Prochlorococcus and Synechococcus was differentially distributed with the former dominant in the tropical Pacific and the latter more abundant in the subtropical and temperate Pacific. They also report that these picocyanobacteria contribute ~45% of the total primary production in the tropical Pacific and ~70% of total in the subtropical and temperate Pacific.

Strengths:

The manuscript is generally well written and employs sound scientific techniques. The rationale for the study is introduced properly and the results adequately described and discussed.

Major Criticism:

The Fouilland et al paper that the authors cite for their method of using Cycloheximide as a protein synthesis inhibitor specific for Eukaryotes, suggests using Cycloheximide in conjuction with Streptomycin (to specifically inhibit Bacteria) to get relative contributions of heterotrophic bacteria and phytoplankton to the NO3−, NH4+ and urea uptake rates. Could the authors explain their rationale to omit using Streptomycin (or other Bacteria specific inhibitors) in this study?

Heterotrophic bacteria have also been shown to contribute substantially to both Carbon fixation (dark) as well as nitrogen fixation. It might be useful to know the estimates of heterotrophic contributions to the overall productivity in the Pacific ocean.

Minor Criticism:

Line 41: Typo; change to “among dominant water masses”

Line 43: Typo; change to “long-term research”

Line 51: Typo; change to “physico-chemical”

Lines 154-155: The fact that temperature and salinity measured for the SP zone was only measured at one station should be mentioned clearly here.

Lines 168-169: Again, the mean euphotic depth for SP is coming only from one station (and hence is not technically a mean value across the region). The authors should be careful in phrasing this clearly here.

Lines 196-202: The abundance values (cells m-2) are not formatted scientifically. The exponential values should be superscripted.

Line 278: Typo; change to “physico-chemical”

Lines 279-281: Grammatically incomplete sentence; Is it supposed to be in continutation with the previous sentence?

Line 287: “typical of temperature oligotrophic waters” Is it supposed to mean “high temperature oligotrophic waters”?

Lines 301-303: Grammatically incorrect; Perhaps the authors meant to write “Based on the hourly carbon uptake rates by total phytoplankton which were estimated in this study”

Line 313: The ± sign is missing in the Nitrate and Ammonium uptake rates for the TP region.

Author Response

Major Criticism:

The Fouilland et al paper that the authors cite for their method of using Cycloheximide as a protein synthesis inhibitor specific for Eukaryotes, suggests using Cycloheximide in conjuction with Streptomycin (to specifically inhibit Bacteria) to get relative contributions of heterotrophic bacteria and phytoplankton to the NO3−, NH4+ and urea uptake rates. Could the authors explain their rationale to omit using Streptomycin (or other Bacteria specific inhibitors) in this study?

=>As you mentioned, Cycloheximide inhibit euphotic phytoplankton whereas Streptomycin inhibits Bacteria. In this study, we designed to measure Bacteria contribution to the total production so that we inhibit euphotic phytoplankton with Cycloheximide. Using Cycloheximide and Streptomycin, Fouilland et al. (2007) tried to prove that each metabolic inhibitor blocks the targeted groups (euphotic phytoplankton vs bacteria) and compared the nitrogen uptake rate of Cycloheximide and Streptomycin with those in control (without any inhibitor), They found that each metabolic inhibitor blocks the targeted groups. Based on this result, we assured that We could measure only bacteria production using Cycloheximide inhibiting euphotic phytoplankton. For a better clarity, we revised in line 133-137, page 4.

Heterotrophic bacteria have also been shown to contribute substantially to both Carbon fixation (dark) as well as nitrogen fixation. It might be useful to know the estimates of heterotrophic contributions to the overall productivity in the Pacific ocean.

=>Based on our dark carbon uptake results from this study, the heterotrophic contributions to the total primary production were 1.5 ± 0.7 % and 8.7 ± 12.8 % for SP and TP, respectively. We addressed this in line 222-224, page 7. 

Minor Criticism:

Line 41: Typo; change to “among dominant water masses”

=> We revised it in line 41.

Line 43: Typo; change to “long-term research”

=> We revised it in line 43.

Line 51: Typo; change to “physico-chemical”

=> We revised it in line 51.

Lines 154-155: The fact that temperature and salinity measured for the SP zone was only measured at one station should be mentioned clearly here.

=> Actually, only one uptake measurement for picocyanobacterial contribution was conducted at the SP zone. For other parameters such as T/S, nutrients, cell abundance and so on, the measurements were conducted at the two stations for the SO zone. For the only one measurement at A89 station, we indicated as SP (A89) in our manuscript. We addressed it in line 132, page 3.

Lines 168-169: Again, the mean euphotic depth for SP is coming only from one station (and hence is not technically a mean value across the region). The authors should be careful in phrasing this clearly here.

=> Please see our comments just above!

Lines 196-202: The abundance values (cells m-2) are not formatted scientifically. The exponential values should be superscripted.

=> We revised them in line 200-203, page 6. We very appreciated with your checking!

Line 278: Typo; change to “physico-chemical”

=> We revised it in line 281, page 11.

Lines 279-281: Grammatically incomplete sentence; Is it supposed to be in continutation with the previous sentence?

=> We revised the sentence in line 283, page 11.

Line 287: “typical of temperature oligotrophic waters” Is it supposed to mean “high temperature oligotrophic waters”?

=> Yes, that is right. We revised it in line 290, page 11.

Lines 301-303: Grammatically incorrect; Perhaps the authors meant to write “Based on the hourly carbon uptake rates by total phytoplankton which were estimated in this study”

=> Yes, that is right. We revised it in line 304, page 11.

Line 313: The ± sign is missing in the Nitrate and Ammonium uptake rates for the TP region.

=> We added it in Table 4.
